# Prognosis and Treatment of Gastric Cancer: A 2024 Update

**DOI:** 10.3390/cancers16091708

**Published:** 2024-04-27

**Authors:** Claudia Burz, Vlad Pop, Ciprian Silaghi, Iulia Lupan, Gabriel Samasca

**Affiliations:** 1Institute of Oncology “Prof. Dr. Ion Chiricuta”, 400015 Cluj-Napoca, Romania; cristina.burz@umfcluj.ro (C.B.); pop.vlad.vasile@elearn.umfcluj.ro (V.P.); 2Department of Immunology, Iuliu Hatieganu University of Medicine and Pharmacy, 400162 Cluj-Napoca, Romania; 3Department of Biochemistry, Iuliu Hatieganu University of Medicine and Pharmacy, 400338 Cluj-Napoca, Romania; silaghi.ciprian@umfcluj.ro; 4Institute for Interdisciplinary Research in Bio-Nano-Sciences, 400271 Cluj-Napoca, Romania; iulia.lupan@ubbcluj.ro; 5Department of Molecular Biology, Babes-Bolyai University, 400084 Cluj-Napoca, Romania

**Keywords:** prognosis, treatment, gastric cancer, update, 2024

## Abstract

**Simple Summary:**

The prevalence of cancer related to the digestive system keeps rising. We examined the 2024 literature on gastric cancer. Apart from surgery, which remains the primary treatment option for gastric cancer, immunotherapy and therapeutic targeting are becoming increasingly significant in the management of this disease. Following gastric cancer surgery, a multidisciplinary approach is required, with nutritionists and psychologists playing fundamental roles.

**Abstract:**

Due to the high death rate associated with gastric cancer, a great deal of research has been conducted on this disease. The goal of this paper was to start a trimestral review of 2024 for the year that had just started. The scientific literature from 1 January 2024 was chosen with consideration of the the guidelines of the European Society of Medical Oncology (ESMO), which are updated with new findings but not systematically reviewed annually. We used the search term “gastric cancer” to find the most current publications in the PubMed database related to the prognosis and treatment of gastric cancer. As previously said, the only articles that satisfied the inclusion criteria were those from 2024. Articles with case reports were eliminated since they had nothing to do with our research. The treatment of gastric cancer is the focus of the majority of articles from 2024. The primary research axes include surgery and immunonutrition, immunotherapy and *Helicobacter pylori*, and therapeutic targets. Patients with GC may experience less psychological, social, and financial hardship if the recently identified markers discovered in circulation are better assessed and validated. This could be achieved by either including the markers in an artificial intelligence-based diagnostic score or by using them in conjunction with traditional diagnostic methods. Due to the rising death rate associated with GC, funding for research into diagnosis, prognosis, therapy, and therapeutic targets is essential.

## 1. Introduction

The incidence of gastrointestinal cancer continues to increase, as does the age-standardized rate at diagnosis. High global variation is present, with the highest rates being detected in South America, Eastern Asia, and Central and Eastern Europe [1,2]. Globally, gastric cancer (GC) is the most common cause of cancer-related death, with the majority occurring within the first year of diagnosis. A significant predisposing factor for GC is gastric intestinal metaplasia (GIM) [3]. GC may coexist with benign disorders such as chronic pseudotumoral pancreatitis or other malignancies such as gallbladder cancer as a part of genetic predisposition (Li Fraumeni and Lynch syndrome, familial adenomatous polyposis) [1,4].

Because there is currently no effective treatment for GC and detection is often delayed, GC is still one of the most serious diseases in the world and has a poor prognosis. Many GC patients are classified as having advanced GC, with extremely short lifespans, as a result of delayed diagnosis [5]. Treatment plans for GC now include a strong emphasis on multidisciplinary team consultation. Nonetheless, a plethora of therapeutic protocols and clinical trial insights exist [6]. The majority of patients with GC have a good prognosis because personalized therapy has been implemented; however, some patients still have advanced GC with distant metastases and recurrence [7].

## 2. Materials and Methods

The aim of the article was to begin a systematic literature review of 2024 as a trimestral work for the year. Scientific literature was selected from 1 January 2024 while also keeping in mind the European Society of Medical Oncology (ESMO) recommendations, which are not reviewed systematically every year as a whole but with small updates being added in line with new studies. We applied the keyword “gastric cancer” to search the PubMed database for the most recent significant articles related to the prognosis and treatment of gastric cancer. As stated before, articles from the year 2024 were the only ones that met the inclusion requirements. Case report articles were excluded because they were not part of our investigation.

## 3. Results

Table 1 shows the number of articles concerning specific research directions relating to gastric cancer.

### 3.1. New Experimental Methods of Diagnosis

From the point of view of experimental approaches, *H. pylori*-infected cytotoxin-associated gene A (CagA)-positive GC cells exhibit the characteristics of cancer stem cells, including enhanced expression of cluster of differentiation (CD) 44, which is a particular surface marker for cancer stem cells. In addition, they present a greater capacity to form tumor spheroids than other GC cell lines [8].

In cases of locally advanced GC, spectral computed tomography (CT) and diffusion-weighted magnetic resonance imaging (DWI) have proven to be helpful in predicting the pathologic response to neoadjuvant chemotherapy (NAC) treatment. Pathologic response was correlated with the quantitative measures of apparent diffusion coefficients and normalized iodine concentration during the delay phase; their combination showed additional benefits and was linked to patient disease-free survival [9]. Moreover, contrast-enhanced computed tomography (CE-CT) imaging can be used to distinguish stomach gastrointestinal cancers (GISTs) from gastric schwannomas (GSs). Delong’s test revealed no statistically significant differences in the prediction performance between the clinical and radiomic data models [10].

In addition, the blood levels of serum aldehyde dehydrogenase 3 family member B1 (ALDH3B1) represent a potential diagnostic biomarker for GC. Its efficacy increases when it is accompanied by the presence of carcinoembryonic antigens (CEAs) [11]. Furthermore, the preoperative C-reactive protein-to-albumin ratio is another biomarker that may have the potential to predict surgical outcomes in terms of both short- and long-term survival [12]. On the other hand, for postoperative recurrence, a useful prognostic marker is serum New York esophageal squamous cell carcinoma 1 (NY-ESO-1) antibodies, which can be detected [13].

Other markers have been evaluated in clinical practice to determine their prognostic value. By using reverse-transcription polymerase chain reaction (qRT–PCR), the levels of cytokeratin 20 (CK20) and transmembrane glycoprotein mucin 1 (MUC1) can be determined from blood samples of GC patients. Tumor size and MUC1 levels were directly correlated with CK20 levels, but the expression and prognostic value of both markers revealed no difference in disease-free survival or overall survival (OS) [14].

Fourier-transform infrared (FTIR) microscopy revealed biomacromolecular alterations during arsenic trioxide (As_2_O_3_)-induced apoptosis in the human gastric adenocarcinoma (AGS) cell line. Therefore, FTIR may be helpful in the study of apoptosis. The apoptotic effects of As_2_O_3_ on AGS cells were validated using flow cytometry data [15].

### 3.2. Prognosis

The OS rate of patients with advanced GC has increased as a result of well-conducted surgery involving lymph node dissection gastrectomy (15 lymph nodes), human epidermal growth factor receptor-2 (HER2), and programmed death-ligand 1 (PD-L1) immunohistochemistry, as well as the approval of new therapeutic lines [1,16,17]. Because gastric muscle fibres are mesodermal tissue, it is hypothesized that increased myogenesis in GC participates in epithelial-to-mesenchymal transition, promoting the metastatic process and poor survival. Thus, after transcriptomic analyses of GC tissue, it was revealed that myogenesis is linked to decreased cell proliferation, which increases epithelial-to-mesenchymal transition, amplifying the process of angiogenesis and proving its poor prognostic value [18].

Additionally, integrin alpha-beta6 (ITGB6) and Rac family small GTPase 1 (Rac1) are markers of poor prognosis and tumor growth in patients with GC. ITGB6 may work by targeting Rac1, as evidenced by in vitro research showing that ITGB6 and Rac1 increase the proliferation, migration, and invasion of GC cells [19]. Furthermore, patients with GC who express higher levels of integrin beta-like 1 (ITGBL1) and lower levels of fibulin-2 (FBLN2) have a negative prognosis. Through the AKT (also known as protein kinase B)/FBLN2 axis, ITGBL1 contributes to the promotion of metastasis and strengthening of GC cell resistance to anoikis [20].

In contrast, the AIO-FLOT3 (Arbeitsgemeinschaft Internistische Onkologie-fluorouracil, leucovorin, oxaliplatin, and docetaxel) trial revealed a good prognosis by prospectively assessing the effectiveness of multimodal treatments for GC patients with oligometastases, such as surgical excision of primary and metastatic lesions in conjunction with chemotherapy [21].

#### 3.2.1. Research Models

The prognosis is determined by several independent factors, including age, surgery, chemotherapy, and the tumor-node-metastasis (TNM) staging system. In light of this, the nomogram is a model that can be used to predict both cancer-specific survival (CSS) and OS in patients with gastric signet cell carcinoma [22] and is facilitated by machine learning (ML) models available for diagnosis. Using a noninvasive method such as a radiomic-clinicopathological model of this kind may accurately predict perineural invasion (PNI) in GC patients before surgery. Patients with PNI who had GC had comparatively poor prognoses [23]. The use of imaging diagnostics is limited because 42.5% of metastatic lymph nodes in GC are of the nodular type or peripheral type. However, ML models provide good predictive power for GC lymph node metastases [24]. A subset of ML known as deep learning (DL) was developed to predict the postoperative GC patient survival rate with accuracy. The DL model outperformed the other ML models in terms of net gains at three years, according to decision curve research [25].

#### 3.2.2. Immunological Indicators

Through type 2 (Th2) immune responses, the immune system is crucial in oncogenesis, from gastritis to metaplasia, dysplasia, and GC. A wide variety of cell types and cytokines are involved in the processes that lead to chronic inflammation and carcinogenesis, and Th2 immune responses are among the main players in this process. Specifically, the process is mediated by the activity of cytokines, such as IL-33 and IL-13, and cell types, such as mast cells, M2 macrophages, and eosinophils, which promote diffuse and chronic gastritis-dependent metaplasia [26].

Several markers have been established as potential prognostic factors in clinical research. One of them is represented by the alpha 11 integrin subunit (ITGA11), a prognostic factor associated with immunity in gastric adenocarcinoma, a factor that is crucial for modelling the tumor immune microenvironment. Both immune cell infiltration and immune checkpoint marker expression are increased in patients with gastric adenocarcinoma who have elevated ITGA11 expression [27]. The PANoptosis (a unique new form of programmed cell death) signature can predict the prognosis and immunological efficacy of GCs in addition to aiding in the identification of the features of the tumor microenvironment. In addition to having a positive prognosis, patients with a low PANoptosis-related risk score (PANS) also had low tumor purity, high microsatellite instability (MSI), high tumor mutation load (TMB), and sensitivity to immunotherapy. The PANS can help determine whether patient demographics are appropriate for a certain chemotherapeutic medication [28]. Elevated levels of CD3+ tumor-infiltrating lymphocytes (TILs) were found to be strongly correlated with better survival rates and may be used as prognostic indicators. Furthermore, Epstein–Barr virus-positive and PD-L1-positive GCs were associated with CD3+ T-cell infiltration, which could help identify targets for immunotherapy [29]. In patients with GC, the immune–metabolism signature, which is linked to the ratio of active CD4+ T cells to regulatory T (Treg) cells, can be used to evaluate treatment plans, the tumor microenvironment, and patient prognosis. When the ratio of activated CD4 T cells to Tregs was less than 1, GC was found to have a predictive adverse effect [30].

### 3.3. Treatment

Table 2 shows the number of articles concerning specific research directions in relation to GC treatments.

#### 3.3.1. Immunotherapy and *Helicobacter pylori* (*H. pylori*)

The presence of an immunological tumor center, which includes tumor-reactive chemokine ligand 13 (CXCL13) T cells, epithelial interferon-stimulated gene programs, and early immune remodelling characterized by enhanced infiltration of CD8+ T cells, is responsible for the clinical response to first-line chemoimmunotherapy for advanced GC [31]. Claudin 11 (CLDN11) and atypical chemokine receptor 3 (ACKR3), traditionally called CXCR7-positive fibroblasts, are crucial for the management of GC with peritoneal metastases [32]. A retrospective cohort study evaluated the role of neoadjuvant chemoimmunotherapy by camrelizumab + nab-paclitaxel + S−1 vs. neoadjuvant chemotherapy alone by nab-paclitaxel in 128 patients with GC. The addition of camrelizumab to chemotherapy may improve pathological response and prolong the first recurrence. Moreover, postoperative complications and side effects associated with combined therapy did not increase compared to those associated with single therapy [33].

Furthermore, the addition of immunotherapy has revealed great efficacy, with manageable levels of toxicity being observed in comparison to conventional treatment [34]. One example is pyroptosis, a proinflammatory programmed cell death mediated by an inflammasome. It has multiple effects, influencing the onset and progression of GC in distinct ways. By stimulating the secondary pyroptosis pathway, controlling the nucleotide-binding oligomerization domain-like (NOD)-like receptor family pyrin domain containing 3 (NLRP3) inflammasome, and blocking caspase-1, several pyroptosis-based treatments have been discovered to prevent GC. Consequently, pyroptosis scores can be utilized to predict the effects of immunotherapy on GC patients [35].

It has been well established that *H. pylori* plays a role in the oncogenesis of GC; thus, it is strongly recommended that all patients with infection be treated. Therefore, current clinical research focuses mostly on developing strategies to prevent GC and developing treatments to combat increasing antibiotic resistance [36]. The presence of Mott cells (plasma cells that contain Russell bodies) is an independent, helpful predictor factor. Laboratory data suggest that the presence of these cells is associated with an early disease stage and a good prognosis. Thus, these cells may play a significant role in the development of *H. pylori* infection-related cancer [37]. The genetic diversity of *H. pylori* varies according to geographical location [38]. Several studies worldwide have attempted to evaluate the rate of success of first-line treatment in the pediatric population. Among the 53 patients who received treatment, the rate of eradication was only 38%. The eradication rate was below expectations, which raises many questions and underlines the importance of developing more potential medical strategies, especially through the study of antimicrobial testing [39].

One interesting point of view is that *H. pylori* pIRES2-DsRed-5Express-ureF DNA vaccination may be useful as an immunotherapeutic treatment in individuals with advanced GC. *H. pylori* DNA vaccines elicit a change in the response from Th1 to Th2, mimicking the immunological conditions of GC patients with persistent *H. pylori* infection. [40].

Additional viable options for the treatment of *H. pylori* infections include antimicrobial peptide (AMP) hydrogels, which exhibit both efficacy and biosafety. Because of their rapid physical membrane disruption and anti-inflammatory/immunoregulatory qualities, AMP have been shown to offer special advantages over antibiotics-resistant microorganisms. AMP hydrogels offer two advantages over traditional antibiotic treatments: they eliminate the need for proton pump inhibitors throughout the treatment and quickly kill germs in the gastric juice [41].

#### 3.3.2. Possible Experimental Treatments from Plant Extracts

Apigenin may act as an anti-GC agent, which is demonstrated by its ability to modulate multiple cancer hallmarks in GC, including cell proliferation, apoptosis, migration, inflammation, and oxidative stress [42]. Reversing stomach precancerous lesions, particularly dysplasia, was safe and effective with one pack of Moluodan, a Chinese herbal medicine, taken three times a day for a year. Greater efficacy was observed when the dose was doubled [43].

By increasing the generation of reactive oxygen species (ROS) in the mitochondria, decreasing the potential of the mitochondrial membrane, and blocking the STAT3 pathway, asarinin causes apoptosis and slows the progression of gastric precancerous lesions [44]. The programmed cell death 4 (PDCD4)-autophagy related 5 (ATG5) signalling pathway may be regulated by the ethyl acetate extract of Celastrus orbiculatus to suppress autophagy in gastric epithelial cells, thereby reversing the progression of precancerous lesions in the GC [45].

Eremia multiocellata, in combination with cisplatin, successfully decreased the migratory and invasive capacity of the GC cell line MKN45/cisplatin DDP and triggered MKN45/DDP cell apoptosis, according to the results of both in vitro and in vivo investigations [46]. With a distinct aniline–indole fused moiety, anithiactin D [1], a novel member of the 2-phenylthiazole class of natural compounds, was isolated from the marine mudflat-derived actinomycete *Streptomyces* sp. 10A085. It is a strong inhibitor of cancer cell motility [47].

Furthermore, silver nanoparticles (AgNPs) produced from the extract of Caralluma pauciflora may be utilized to treat GC in humans [48]. Portulaca oleracea L. (POL) may be a viable new option for the management and prevention of digestive system malignancies associated with inflammation. POL has various chemical formulas that include organic acids, terpenoids, alkaloids, flavonoids, and other types of natural substances [49].

#### 3.3.3. Chemotherapy

New therapeutic options are emerging every year, as in the case of advanced PD-L1-positive GC patients with extensive peritoneal metastases, where the combination of nivolumab plus modified oxaliplatin (L-OHP) with l-leucovorin (l-LV) and a bolus or continuous infusion of 5-fluorouracil (5-FU) (mFOLFOX6) has been proven to be safe and have moderate effectiveness [50]. In patients with metastatic GC, the second-line therapy, programmed cell death protein 1 (PD-1) inhibitor (selected according to patients’ requirements) in combination with albumin paclitaxel (125 mg/m^2^, intravenously, days 1 and 8, or 250 mg/m^2^, intravenously, day 1) and apatinib (250 or 500 mg, orally, days 1–21) every 3 weeks, has demonstrated modest efficacy and safety [51]. Chemotherapy plus a cell division control protein 42 (Cdc42) inhibitor has demonstrated encouraging antitumor effectiveness in patients with resistant GC HER2+, offering valuable information for developing treatment plans for patients with trastuzumab-resistant GC [52]. By increasing the expression of inducers of drug sensitivity markers, such as those in the serum amyloid A (SAA) family and suppressing the expression of identified drug resistance marker suppressors, such as those in the melanoma antigen (MAGE) family, trefoil factor (TFF) family, and Ras-associated binding 25 (RAB25), it may be possible to decrease drug resistance and improve the efficacy of chemotherapy for GC [53]. According to in vivo investigations, the injection of F. nucleatum extracellular vesicles (Fn-EVs) increased not only the growth of GC tumors and their metastasis to the liver but also GC-induced tumor resistance to oxaliplatin [54].

#### 3.3.4. Surgery and Immunonutrition

In GC surgery fulfils not only e treatment role, but also an important staging one alongside with the CT scan, endoscopic ultrasound. For stages IB-III eligible for complete respectability a laparoscopy with peritoneal washings for cancer cells is recommended in order to eliminate potential macroscopic or radiologically occult peritoneal metastases [1].

Surgical resection, as a part of the multimodality treatment in GC, remains the keystone for achieving a curative state of disease for stage IB-III disease. It can be realized via endoscopic resection only in early gastric cancers that fulfill the exact criteria for eligibility: well-differentiated G1-2, T1a confined to the mucosa, and non-ulcerated tumors. In clinical practice, two endoscopic modalities are approved: endoscopic mucosal resection for lesions < 10–15 mm and endoscopic submucosal dissection for superficial lesions [1].

Neoplastic lesions that are not eligible for endoscopic resection, along with IB-III stages, are imperatively approached by surgery, which can be realized through the use of laparotomy, laparoscopy, and robotic surgery. Laparoscopic surgery, the standard of care, has been evaluated as being non-inferior to laparotomy, in addition to presenting a low morbidity and shorter recovery time [1]. In terms of short/long-term outcomes, one Japanese center attempted to respond to this question by conducting a retrospective study with patients undergoing radical laparoscopy or open total gastrectomy surgery between 2010 and 2020. By choosing the primary outcome of relapse-free survival, it has been revealed that laparoscopic total gastrectomy improved the short and long-term outcomes in early stages, while this was true in the case of open gastrectomy for advanced cases of GC [55].

There are numerous psychosocial effects of surgery on patients with GC, especially preventive complete gastrectomy for hereditary GC. Moreover, these symptoms are closely linked to physical symptoms, indicating the importance of counselling patients to make informed decisions under the surveillance of a multidisciplinary team [56]. In regard to postoperative weight reduction and overall quality of life, particularly dyspepsia, proximal gastrectomy is generally preferable to total gastrectomy [57]. After gastrectomy, patients often return to their presurgery quality of life one year later [58]. Following proximal gastrectomy, reflux symptoms have been linked to relaxation pressure and lower esophageal sphincter residual pressure integrated into high-resolution impedance manometry [59].

Even following neoadjuvant chemotherapy, the risk of metastasis remains significant due to the importance of distal lymph nodes, especially for patients with proximal locally advanced GC (cT3/4 before chemotherapy). Thus, the standard of care remains total gastrectomy [60].

Correct lymphadenectomy diminishes the risk of disease recurrence. For the optimal visualization of lymphatic flow, the use of fluorescent lymphography with indocyanine green (IG) is encouraged. As surgery takes place in multimodal treatments, being carried out after neoadjuvant chemotherapy, the matter of IG was analyzed in this population. Not only did IG increase the mean of the lymph nodes retrieved, but it also diminished the nodal non-compliance ratio [61]. Similar results were confirmed in patients with BMI ≥ 25 kg/m^2^ [62].

The oral–gastrointestinal microbiota axis of GC patients can be greatly impacted by anti-GC treatments, such as chemotherapy and gastrectomy, while on the one hand, the microbiota and its metabolites can have a major influence on the course of the disease [63]. A novel theoretical foundation for cancer management against GC is the restoration of intestinal microbial butyrate, which increases CD8+ T-cell cytotoxicity through hydroxycarboxylic acid receptor 2 (HCA2), also known as GPR109A/HOP homeobox (HOPX) and inhibits GC carcinogenesis [64].

Following major GC surgery, the administration of adjuvant chemotherapy, in addition to tumor-specific total nutrients, significantly enhances the patient’s immune system and nutritional status [65]. Immunonutrition decreased postoperative complications, shortened hospital stays, and improved nutritional outcomes in a group of 55 patients given formulas that included arginine, nucleotides, omega-3 fatty acids, and extra virgin olive oil [66].

The cornerstone of curative care is surgical resection, although the best surgical approach is still under debate. In total, 98% of adverse events were classified as “serious”, 85% of tumor removals were classified as complete, and 74% of surgery-related deaths were reported. The literature does a poor job of reporting critically key outcomes, and subsequent trials have not shown any improvement in the situation. It will take more effort to increase uptake [67].

However, another possibility for radical gastrectomy is represented by the robotic approach. In overweight patients (BMI ≥ 25 kg/m^2^) diagnosed with GC, the role of this surgery remains controversial. A retrospective analysis of 482 cases that fulfilled the inclusion criteria between 08/2016 and 12/2019 demonstrated that the robotic approach from the point of view of lymphadenectomy, postoperative rehabilitation, and clinical outcomes are comparable with the laparoscopic approach for this category of patients [68]. Several surgical robots have been developed, even if the da VinciTM Surgical System is the one that is utilized the most frequently worldwide. In November 2022, the Japanese HinotoriTM Surgical Robot System was introduced for clinical use in Japan. With the Hinotori surgical robot, safe robotic gastrectomy combined with regular lymphadenectomy for GC can be carried out [69]. In the context of GC surgery, the textbook outcome (TO) is a multifaceted indicator of surgical quality. The application of TO in laparoscopic gastrectomy establishes a standard for attaining better prognoses and enables surgeons to devise methods to enhance surgical care [70].

In advanced GC patients with peritoneal carcinomatosis, hyperthermic intraperitoneal chemotherapy (HIPEC) was assessed in the CYTO-CHIP study. The HIPEC group was compared to the surgery-only control group, with the results suggesting that HIPEC improves OS along with recurrence-free status without exacerbating morbidity and mortality. Nonetheless, the value of the median peritoneal cancer index (PCI) persisted at a high value in the HIPEC group after the treatment, suggesting the effectiveness of cytoreductive surgery combined with HIPEC in patients with a low PCI score. The NCCN guidelines consider HIPEC to be an alternative only in selected patients with peritoneal carcinomatosis [71].

In the metastatic state, total gastrectomy is not recommended by the guidelines in order to improve survival. Its use is only recommended with palliative intent due to the presence of symptomatic disease, which is considered refractory to conservative treatments [72]. A retrospective study that evaluated stage IV GC patients who underwent surgery for palliative procedures (total/subtotal gastrectomy or gastric bypass) describes that in terms of survival and perioperative morbidity, gastrectomy was preferred over bypass [73].

#### 3.3.5. Therapeutic Targets

Possible therapeutic targets are being researched. One of them, prefoldin subunit 5 (PFDN5), may influence the immune system, cell cycle, and cell death to facilitate the growth of GC cells. These discoveries offer a new understanding of the precise therapies and molecular mechanisms involved in GC [74].

By controlling the nuclear factor of activated T cells (NFAT1) and interferon regulatory factor 1 (IRF1) pathways, elevated insulin-like growth factor 2 mRNA-binding protein 3 (IGF2BP3) promotes the progression of GC both in vivo and in vitro. Targeting IGF2BP3 may be a viable therapeutic strategy for the treatment of GC [75].

Apatinib resistance is mediated by YY1 transcription factor (YY1) overexpression, which also inhibits ferroptosis in GC cells and reduces immune cell infiltration in GC tumors via the p53 pathway. As a result, YY1 is a prospective therapeutic target for improving patient outcomes since it plays a crucial role in GC progression and prognosis [76].

Tyrosine 3-monooxygenase/tryptophan 5-monooxygenase, or YWHAZ, operates in tandem with transmembrane protein 65 (TMEM65) to activate PI3K-Akt-mTOR (an intracellular signalling pathway important in regulating the cell cycle) signalling, which in turn promotes gastric carcinogenesis. In GC, the overexpression of TMEM65 is a therapeutic target and could function as a potential independent biomarker [77].

Enhanced Zeste homologue-2 (EZH-2)-interacting lncRNAs are intriguing candidates for the development of new targeted therapeutics for GC and have a role in gastric carcinogenesis. Many tumor suppressor genes include promoter regions where EZH2 can be recruited by EZH2-interacting lncRNAs, which then use histone methylation to deactivate the transcription of those genes. When EZH2 and this lncRNA interact, various processes, including drug resistance, migration, invasion, metastasis, and the cell cycle, are modulated in in vitro and in vivo GC models [78].

The primary cause of disease progression is the heterogeneity of GC. Tumor protein (TP)53 mutations were more prevalent in patients whose tumors were HER2-positive. Mutations in KRAS Proto-Oncogene GTPase (KRAS), TP53, Phosphatidylinositol-4,5-Bisphosphate 3-Kinase Catalytic Subunit Alpha (PIK3CA), phosphatase and tensin homologue (PTEN), and Erb-B2 Receptor Tyrosine Kinase 2 (ERBB2) cause an increase in PD-1 expression in PD-L1-positive tumors. Changes in the variables linked to neoangiogenesis are linked to TP53 and PTEN mutations. The patients who had the highest response to treatment, including complete morphologic regression, were those without aggressive growth indicators, as confirmed by an examination of the molecular characteristics [79]. The frequency of mutations was substantially greater in advanced GC patients than in GC patients. Advanced GC patients showed a greater frequency of cumulative genetic events, such as increased rates of PIK3CA mutations, better detection of immunotherapy biomarkers, and mutations of the estrogen receptor 1 (ESR1) gene, in addition to modifications of PIK3CA, KRAS, and ERBB2 as somatic oncogenic drivers [80].

As prospective therapeutic targets for patients with refractory GC receiving cisplatin-fluorouracil (CF) combination chemotherapy, the biomarkers Centromere Protein B (CENPB), Metastasis-Associated 1 (MTA1), Glucosaminyl (N-Acetyl) Transferase 3, and Mucin Type (GCNT3) are promising in terms of clinical use. In patients with CF-resistant GC, the mRNA and protein expression levels of CENPB, MTA1, and GCNT3 are elevated, which is strongly linked to poor OS [81].

Since circRNAs play crucial roles as noncoding RNAs (ncRNAs) in GC, circular RNAs (circRNAs) have been the subject of much research. Highly expressed circular RNA_0023685 may be a novel clinical diagnostic biomarker and therapeutic target for GC, and it may also promote GC genesis [82]. Because of the mechanism by which elevated p62 expression, a classical receptor of autophagy, accelerates the progression of GC, the p62 protein represents an additional intriguing therapeutic target for antitumor treatment in relation to GC patients. Compared to that seen in normal tissues, the expression of p62 in GC tissues was notably greater. In people with GC, the expression of p62 was positively linked with a poor prognosis. According to in vitro cell tests, p62 stimulates the migration and proliferation of GC cells. Mechanistically, increased p62 expression causes epithelial–mesenchymal transition (EMT), which in turn leads to the downregulation of vimentin and N-cadherin and the overexpression of E-cadherin [83].

Disseminated intravascular coagulation (DIC) and bone marrow metastases are two characteristics of highly aggressive gastric cancer (HAGC). In HAGC peripheral blood mononuclear cells (PBMCs), a significant increase in the number of neutrophils was observed. Despite being immature, these neutrophils are activated, these neutrophils are active. T-cell production was inhibited and decreased, and mononuclear phagocytes displayed M2 macrophage-like characteristics. Cell–cell crosstalk analysis revealed that HAGC enhanced the expression of various signalling pathways, including APP-CD74, MIF-(CD74+CXCR2), and MIF-(CD74+CD44), which are implicated in the inhibition of T-cells by neutrophils. Additionally, HAGC showed the upregulation of NOTCH signalling, vascular endothelial growth factor A (VEGF), platelet-derived growth factor subunit B (PDGF), fibroblast growth factor 2 (FGF), and the NETosis-associated genes S100 calcium-binding protein A8 (S100A8) and S100 calcium-binding protein A9 (S100A9), all of which are involved in the development of DIC. Since highly aggressive GC (HAGC) neutrophils have high expression levels of S100A8 and S100A9, these proteins may be used as novel targets for HAGC diagnosis and treatment [84].

Claudin 18.2 (CLDN18.2), a member of the tight junction protein family that may act as a potential therapeutic target, is expressed in gastric adenocarcinoma. It has been observed that 14–87% of gastric adenocarcinoma patients express CLDN18.2. Since CLDN18.2 is expressed on the outer cell membrane, it is accessible for binding with monoclonal antibodies (mAbs) [85].

## 4. Future Directions

In addition to the most frequently used imaging and endoscopic techniques to diagnose GC, it is necessary to ascertain the nature of GC at an early stage. For this, we can detect circulating tumor cells, as well as circulating cell-free tumor DNA, by using a “liquid biopsy”, an innovative screening method applied in other types of cancer [86]. On the other hand, according to Reppeto et al. [87], there are new barriers to overcome regarding the circulating proteomic biomarkers found in blood and other fluids in GC, such as altered glycosylation signatures of circulating proteins or circulating exosomal proteins. However, no single protein was able to serve as an adequate marker, but a metabolomic ML predictor was recently invented and tested for the diagnosis and prognosis of GC [88].

## 5. Conclusions

An improved assessment and validation of the newly discovered markers found in circulation, either by their inclusion in a diagnostic score using artificial intelligence or in combination with classic diagnostic techniques, could mitigate the psychological, social, and economic burden of patients with GC. Consequently, the availability of funding for research in the area of diagnosis, prognosis, treatment, and therapeutic targets for GC is crucial due to the increasingly high mortality rate of this disease.

## Figures and Tables

**Table 1 cancers-16-01708-t001:** Literature analysis for the year 2024.

	Found Articles (Number)
New Experimental Methods of Diagnosis	8
Prognosis	15
Treatment	47

**Table 2 cancers-16-01708-t002:** Literature analysis for the year 2024.

	Found Articles (Number)
-Immunotherapy and *H. pylori*	11
-Plant Extracts	8
-Chemotherapy	5
-Surgery and Immunonutrition	19
-Therapeutic Targets	12

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
