# Peer review of "Prognosis and Treatment of Gastric Cancer: A 2024 Update"

_cancers, 2024, doi:10.3390/cancers16091708_

Round 1

Reviewer 1 Report

Comments and Suggestions for Authors

I read the manuscript carefully.

The authors aim for an unrealistic result, considering the fact that 2024 began a few months ago.

I cannot understand whether the manuscript is a narrative review or a systematic literature review.

The Abstract is missing.

A paragraph relating to materials and methods is missing to find out how the articles in the literature were selected.

The references to surgical treatment are not very precise. The minimum number of lymph nodes to be removed for a correct lymphadenectomy is at least 16 and not 15 as written.

The mention of laparoscopic and robotic surgery is minimal. No mention is made of the use of indocyanine green for evaluating the extent of lymphadenectomy, especially in early gastric cancer. Likewise, no reference is made to endoscopic resections in the treatment of early gastric cancer.

In the case of advanced gastric cancer with peritoneal carcinomatosis, no reference is made to peritonectomy with HIPEC. Nothing on palliative surgery in gastric cancer with complications.

Author Response

Reviewer 1.

I read the manuscript carefully.

  1. The authors aim for an unrealistic result, considering the fact that 2024 began a few months ago.

The aim of the article was to begin a review of 2024 as a trimestral work for the year which began. Scientific literature was selected from 01/01/2024, but keeping in mind the European Society of Medical Oncology (ESMO) recommendations which are not reviewed systematically every year as a whole, but with small updates with the new studies.  

  1. I cannot understand whether the manuscript is a narrative review or a systematic literature review.
  • Our article is a systematic literature review.

  1. The Abstract is missing.
  • We added

  1. A paragraph relating to materials and methods is missing to find out how the articles in the literature were selected.

  • We added

  1. The references to surgical treatment are not very precise. The minimum number of lymph nodes to be removed for a correct lymphadenectomy is at least 16 and not 15 as written.

As practicing oncology in European Union, the standard of care is dictated by the ESMO Guidelines in which it is stated that the “The current AJCC/UICC TNM (8th edition) classification recommends excision of a minimum of 15 lymph nodes for reliable staging. “regarding the D1 gastric resection. We modified the text by adding the word “minimum” and also by explaining furthermore that “In Asian medical practice, the superiority of D2 resection has been demonstrated by randomized and observational trials, leading in improved outcomes on contrast with D1 resection

  1. The mention of laparoscopic and robotic surgery is minimal. No mention is made of the use of indocyanine green for evaluating the extent of lymphadenectomy, especially in early gastric cancer. Likewise, no reference is made to endoscopic resections in the treatment of early gastric cancer.

We added this information:

In GC surgery fulfils not only e treatment role, but also an important staging one alongside with the CT scan, endoscopic ultrasound. For stages IB-III eligible for complete respectability a laparoscopy with peritoneal washings for cancer cells is recommended in order to eliminate a potential macroscopic or radiologically occult peritoneal metastases (1).

Surgical resection as a part of the multimodality treatment in GC remains the keystone for achieving a curative state of disease for stage >IB disease. It can be realized by endoscopic resection only in early gastric cancers which fulfill exact criteria for eligibility: well differentiated G1-2, T1a confined to the mucosa and non-ulcerated tumors. In clinical practice two endoscopic modalities are approved, endoscopic mucosal resection for lesions < 10-15mm and endoscopic submucosal dissection for the superficial lesions (1).

Neoplastic lesions which are not eligible for endoscopic resection along with IB-III stages are imperatively approached by surgery which can be realized by laparotomy, laparoscopy and robotic surgery. Laparoscopic surgery, the standard of care, has been evaluated as non inferior to laparotomy, in addition presenting a low morbidity and shorten recovery time (1). In terms of short/long term outcome, one Japanese center attempted to respond at this question by realizing a retrospective study with patients undergoing radical surgery between 2010-2020 by laparoscopic ore opened total gastrectomy. By choosing the primary outcome the relapse-free survival, it has been revealed that laparoscopic total gastrectomy improved the short and long-term outcomes in early stages, while opened gastrectomy indicated for advanced cases of GC (https://pubmed.ncbi.nlm.nih.gov/38537974/).  

However, another possibility for radical gastrectomy is represented by the robotic approach. In overweight patients (BMI≥25kg/m2) diagnosed with GC the role of this surgery remains controversial. A retrospective analysis of 482 cases which fulfilled the inclusion criteria between 08/2016 and 12/2019 demonstrated that the robotic approach from the point of view of lymphadenectomy, postoperative rehabilitation and clinical outcomes are comparable with the laparoscopic approach for this category of patients (https://pubmed.ncbi.nlm.nih.gov/38627257/). 

A correct lymphadenectomy diminishes the disease risk of recurrence. For an optimal visualization of the lymphatic flow the use of fluorescent lymphography with indocyanine green (IG) is encouraged. As surgery takes part in the multimodal treatment, being underwent after neoadjuvant chemotherapy the matter of IG was analyzed in this population.  Not only that IG increased the mean of the lymph nodes retrieved, but also diminished the nodal non-compliance ratio (https://pubmed.ncbi.nlm.nih.gov/38375670/). Similar results were confirmed in patients with BMI ≥25kg/m2 (https://pubmed.ncbi.nlm.nih.gov/38502275/).

  1. In the case of advanced gastric cancer with peritoneal carcinomatosis, no reference is made to peritonectomy with HIPEC. Nothing on palliative surgery in gastric cancer with complications.

We added the information:

In advanced GC patients with peritoneal carcinomatosis hyperthermic intraperitoneal chemotherapy (HIPEC) was assessed in the CYTO-CHIP study. The HIPEC group was compared to the surgery only control group suggesting that HIPEC improves the OS along with recurrence-free without exacerbating morbidity and mortality. Nonetheless, the value of the median peritoneal cancer index (PCI) persisted at a high value in the HIPEC group after the treatment, suggesting the effectiveness of cytoreductive surgery combined with HIPEC in patients with a low PCI score. The NCCN guidelines consider HIPEC as an alternative only in selected patients with peritoneal carcinomatosis. (https://www.nccn.org/professionals/physician_gls/pdf/gastric.pdf). 

In the metastatic state the total gastrectomy is not recommended by the guidelines in order to improve survival, but only as a palliative intent due to the symptomatic disease which is considered as refractory to the conservative treatment (https://pubmed.ncbi.nlm.nih.gov/38534942/). A retrospective study which evaluated stage IV GC patients which underwent surgery for palliative procedures (total/subtotal gastrectomy or gastric bypass) describes that in terms of survival and perioperative morbidity the gastrectomy was preferred over bypass (https://www.scielo.br/j/abcd/a/td3cM4gNXPFbF5wHDbRCM7p/?lang=en). 

Reviewer 2 Report

Comments and Suggestions for Authors

This is a review of the articles related to gastric cancer published in 2024 and indexed in pubMed database.

Authors summarize the results of 81 publications, providing an overview of several different aspects of gastric cancer treatment, both clinical and experimental.

Comments:

Authors should better explain how they selected the articles commented in this review and specify the time period (January 2024 to…) or when (date) the search was applied.

(Infact, at present, 16 april, the search using the specified criteria provides over 2000 publications…)

Page 4 lines 169-170  …”these genes may play a significant role in the development of H. pylori infection-related cancer”.

Which are these genes? Please explain better this phrase.

In page 7, the results of ref.76 were reported incorrectly. I suggest to make the following changes:

line 331:  “Despite being young, these neutrophils are active”

Change to: Despite being immature, these neutrophils are activated.

line 335: “inhibition of neutrophils by T cells”.

Change to: inhibition of T-cells by neutrophils.

The study reported in ref 78 is similar to ref 72. Therefore, I would suggest to comment ref 78 just subsequently to ref 72.

Comments on the Quality of English Language

Minor editing required

Author Response

Reviewer 2.

  1. Authors should better explain how they selected the articles commented in this review and specify the time period (January 2024 to…) or when (date) the search was applied.

The aim of the article was to begin a review of 2024 as a trimestral work for the year which began. Scientific literature was selected from 01/01/2024, but keeping in mind the European Society of Medical Oncology (ESMO) recommendations which are not reviewed systematically every year as a whole, but with small updates with the new studies.  

We applied the keyword "gastric cancer" to search the PubMed database for the most recent significant articles related to prognosis and treatment of gastric cancer.

  1. Page 4 lines 169-170 …”these genes may play a significant role in the development of H. pylori infection-related cancer”.

Which are these genes? Please explain better this phrase.

Correction made: “cells “ in the place of genes.

  1. In page 7, the results of ref.76 were reported incorrectly.

  1. I suggest to make the following changes:

line 331:  “Despite being young, these neutrophils are active”

Change to: Despite being immature, these neutrophils are activated.

Changed to: Despite being immature, these neutrophils are activated

line 335: “inhibition of neutrophils by T cells”.

Change to: inhibition of T-cells by neutrophils.

Changed to: inhibition of T-cells by neutrophils.

  1. The study reported in ref 78 is similar to ref 72. Therefore, I would suggest to comment ref 78 just subsequently to ref 72.
  • We did what you asked

Round 2

Reviewer 1 Report

Comments and Suggestions for Authors

The authors have supplemented the original manuscript.

I believe that these additions have improved the manuscript.

What the authors write is acceptable.